

# Contributions of Latin American researchers in the understanding of the novel coronavirus outbreak: a literature review

Karen Y. Fiesco-Sepúlveda[1,*] and Luis Miguel Serrano-Bermúdez[2,*]

[1] Epidemiology Program, Faculty of Health Sciences, Universidad Surcolombiana, Neiva, Colombia
[2] Bioprocesses and Bioprospecting Group, Universidad Nacional de Colombia, Bogotá D.C., Colombia
* These authors contributed equally to this work.

## ABSTRACT

This article aimed to give the visibility of Latin American researchers' contributions to the comprehension of COVID-19; our method was a literature review. Currently, the world is facing a health and socioeconomic crisis caused by the novel coronavirus, SARS-CoV-2, and its disease COVID-19. Therefore, in less than 4 months, researchers have published a significant number of articles related to this novel virus. For instance, a search focused on the Scopus database on 10 April 2020, showed 1,224 documents published by authors with 1,797 affiliations from 80 countries. A total of 25.4%, 24.0% and 12.6% of these national affiliations were from China, Europe and the USA, respectively, making these regions leaders in COVID-19 research. In the case of Latin America, on 10 April 2020, we searched different databases, such as Scopus, PubMed and Web of Science, finding that the contribution of this region was 2.7 ± 0.6% of the total publications found. In other words, we found 153 publications related to COVID-19 with at least one Latin American researcher. We summarized and processed the information from these 153 publications, finding active participation in topics like medical, social and environmental considerations, bioinformatics and epidemiology.

## INTRODUCTION

Severe acute respiratory syndrome coronavirus 2 (SARS-CoV-2) is a novel virus that mainly affects the respiratory system through the new coronavirus disease 2019 (COVID-19) (*Ciotti et al., 2020*). COVID-19 has spread quickly; thus, on 11 March 2020, the World Health Organization (WHO) declared it as a pandemic (*Hussain, Bhowmik & Do Vale Moreira, 2020*). Since the first cases reported in Wuhan, China, in December 2019, until 20 April 2020, SARS-CoV-2 has affected most countries in the world, with nearly 2.5 million people infected and 170,000 deaths (*Johns Hopkins University, 2020*). Latin America is no stranger to this reality, totaling approximately 100,000 cases and

Corresponding author
Luis Miguel Serrano-Bermúdez,
lmserranob@unal.edu.co

**Table 1 Summary of the COVID-19 outbreak affectation in the Latin American countries.** Total cases and total deaths are reported until 20 April 2020 (*Johns Hopkins University, 2020*).

| Country | First case | Cases | Deaths |
|---|---|---|---|
| Argentina | Mar 3 | 2,941 | 136 |
| Bolivia | Mar 10 | 564 | 33 |
| Brazil | Feb 25 | 40,581 | 2,845 |
| Chile | Mar 3 | 10,507 | 139 |
| Colombia | Mar 6 | 3,963 | 189 |
| Costa Rica | Mar 6 | 662 | 6 |
| Cuba | Mar 11 | 1,087 | 36 |
| Dominican Republic | Mar 1 | 4,964 | 235 |
| Ecuador | Feb 14 | 10,128 | 507 |
| El Salvador | Mar 18 | 218 | 7 |
| Guatemala | Mar 13 | 289 | 7 |
| Haiti | Mar 2 | 47 | 3 |
| Honduras | Mar 11 | 477 | 46 |
| Mexico | Feb 27 | 8,261 | 686 |
| Nicaragua | Mar 18 | 10 | 2 |
| Panama | Mar 8 | 4,467 | 126 |
| Paraguay | Mar 7 | 208 | 8 |
| Peru | Mar 6 | 16,325 | 445 |
| Uruguay | Mar 13 | 528 | 10 |
| Venezuela | Mar 13 | 256 | 9 |
| Total | | 106,483 | 5,475 |

5,500 deaths, as shown in Table 1 (*Johns Hopkins University, 2020*). However, these data may be lower than actual numbers because the number of tests per million inhabitants remains low, which is caused by factors such as the limited availability of tests and the difficulty of monitoring people without facilities like indigenous populations, vulnerable groups and Venezuelan refugees (*Oliveira, Abranches & Lana, 2020*; *Torres & Sacoto, 2020*).

SARS-CoV-2, as SARS and MERS, belongs to the family *Coronaviridae*, has a zoonotic origin, and can remain on some surfaces for considerable periods (*Ciotti et al., 2020*; *Van Doremalen et al., 2020*). Additionally, COVID-19 is a new disease with no yet vaccines or targeted drugs, making the containment of the outbreak difficult (*Carnero Contentti & Correa, 2020*). Therefore, the recommendation is the self-isolation to reduce COVID-19 spreading, especially in more susceptible people as older adults or patients with comorbidities (*Diaz-Quijano, Rodriguez-Morales & Waldman, 2020*). More general aspects of the current outbreak have been published in review articles according to available information in the moment of publication. These reviews include other zoonotic diseases (such as SARS and MERS), outbreak chronology, virus characteristics, zoonotic links, transmission, diagnosis, disease characteristics, therapeutics and treatments, prevention, epidemiological surveillance and control (*Ciotti et al., 2020*; *Cupertino et al., 2020*; *Huang et al., 2020*; *Millán-Oñate et al., 2020*; *Palacios Cruz et al., 2020*;

*Rodriguez-Morales et al., 2020a*; *Sifuentes-Rodriguez & Palacios-Reyes, 2020*; *Siordia, 2020*; *Wu et al., 2020*; *Zhu et al., 2020*).

In the past, during SARS and MERS outbreaks, research focused on coronaviruses increased significantly, which was led by researchers from China and the USA (*Bonilla-Aldana et al., 2020b*). This new outbreak is not an exception because thousands of articles have been published in less than four months, where China, Europe and the USA are leaders in the number of publications. In the case of Latin America, it is a region with an increasingly high contribution to science; thus, our question was, what are the contributions of Latin American researchers in understanding this novel coronavirus outbreak? Therefore, our purpose in this review was to highlight the contributions of this region in the comprehension of SARS-CoV-2 and COVID-19. The literature survey consisted of revising and summarizing publications with Latin American researchers. Keeping in mind that several researchers from this region work together with researchers from other continents, we included publications submitted by these types of international research groups. Hence, the relevance of this review focused on finding the research interests of Latin American researchers according to global and regional priorities.

## SURVEY METHODOLOGY

### Search strategy

We performed the present review following the PRISMA guidelines. The search was done on 10 April 2020, using Scopus, Web of Science, PubMed, ScienceDirect, Wiley, SAGE, LILACS and SciELO databases because they are the main academic literature collections globally and regionally. Other databases like Springer Link were excluded because they do not allow to filtrate by affiliation. The search equation used had ("COVID 19" OR "COVID-19" OR "SARS-CoV-2" OR "SARS CoV 2" OR "SARS-CoV 2" OR "2019-nCoV" OR "2019 nCoV" OR "nCoV-2019" OR "nCoV 2019" OR "hCoV-19" OR "hCoV 19") in all fields and (Argentina OR Bolivia OR Brasil OR Brazil OR Chile OR Colombia OR Cuba OR Ecuador OR Salvador OR Guatemala OR Haiti OR Honduras OR Mexico OR Nicaragua OR Panama OR Paraguay OR Peru OR Dominicana OR Uruguay OR Venezuela) in affiliation field. We did not consider preprints during the search stage. No interfaces were used in the present literature review.

### Article selection and data extraction

After the search stage, both reviewers (KYFS and LMSB) removed all duplicated publications, which included a manual revision because some publications were simultaneously in English, Spanish, or Portuguese. Later, we performed a second manual revision to verify that all publications had at least one researcher with a Latin American affiliation. After these two manual revisions, we did not exclude more publications, and final publications were included in the qualitative synthesis. Before the qualitative synthesis, we collected the following information, which was used in the bibliometric analysis: title, authors, journal, DOI, type of publication, national affiliation of Latin American researchers, and topic of publication.

## Data analysis

We summarized information from the collected publications according to the type of publication, the topic of publication and the national affiliation in the "Bibliometric Analysis" section. The first purpose of this section was to quantify contributions of the region in the global context, and the contribution by country in the regional context. The second purpose was to classify the publications by topic and type, which allowed us to organize the next sections of this literature review. In the following two sections, "Phylogenetic and Molecular Understanding" and "Medical Contributions", we compiled information from research articles and reviews. The last section, called "Additional Concerns", was included to highlight contributions not covered in the two main topics, but discussed in the remaining publications (commentaries, letters to the editor, editorials, communications, perspectives, points of view and contributions).

## BIBLIOMETRIC ANALYSIS

Following the PRISMA guidelines shown in Fig. 1, we found 301 publications in the considered databases; this number decreased to 161 after excluding duplicates. Later, we manually excluded eight additional publications due to affiliations from New Mexico (1) and Pennsylvania (7) were confused with Mexico and Panama, respectively. Therefore, this qualitative analysis included 153 publications, which contained at least one researcher with Latin American affiliation (see File S1 for complete information of publications). We highlight that several publications were not exclusively submitted by Latin American researchers, some of which are part of research groups together with North American, European, or Asian researchers. Figure 2 presents the classification of publications by type, where most of them were letters to the editor or commentaries, editorials and research articles.

Regardless of the number of authors from the same country but different institutions, among all publications, we highlight that 15 of them were submitted by groups in which there were researchers from at least two Latin American countries. Therefore, these publications were counted for each nation involved; thus, these 15 publications have 56 national affiliations. Most of these publications were due to the Latin American Network of Coronavirus Disease 2019-COVID-19 Research (LANCOVID-19), which was created to integrate the region around this new outbreak (*Rodriguez-Morales et al., 2020c*). The remaining 138 publications were submitted by groups in which there were researchers from a single Latin American country. Therefore, these publications were counted once for the country, disregarding whether they were submitted by one or more researchers with the same national affiliation; in other words, these 138 publications have 138 national affiliations. In summary, the 153 publications accounted for 194 national affiliations. Figure 3 shows publications by national affiliation, where Brazil had the highest contribution with 80 publications, followed by Colombia, Mexico and Argentina, with 36, 18 and 14 publications, respectively. Conversely, the following Latin American countries had no publications: Cuba, Costa Rica, Dominican Republic, El Salvador, Guatemala, Haiti and Nicaragua.

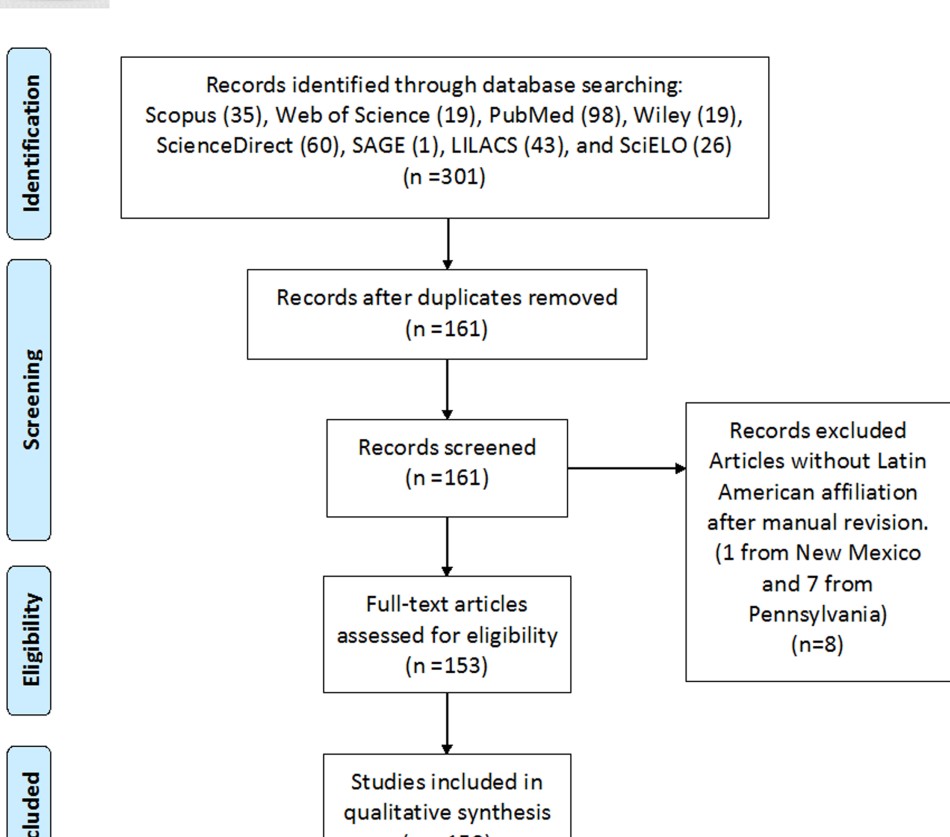

**PRISMA 2009 Flow Diagram**

**Figure 1 PRISMA Flow diagram of selection process of COVID-19 or SAR-CoV-2 publications containing researchers with Latin American affiliation.** Identification stage was performed on 10 April 2020.

We did the same search without the affiliation field restriction. We found 1,224, 615, 3,538, 1,841, 665, 48, 2,627 and 34 publications in Scopus, Web of Science, PubMed, ScienceDirect, Wiley, SAGE, LILACS and SciELO databases, respectively. Hence, publications with Latin American researchers in these databases, Fig. 1, represent 2.9%, 3.1%, 2.8%, 3.3%, 2.9%, 2.1%, 1.6% and 76.5% of all publications, respectively. Excluding SciELO, which is a Latin American database, the contribution of the region was $2.7 \pm 0.6\%$. This low value could be associated with the science gap (gap in science funding, technology, facilities) between the region and the developed countries. However, other possibilities are the late coronavirus appearance in the region (between February and March), as opposed to the initial outbreak (December 2019) and the number of Latin American cases (nearly 4% world total), as shown in Table 1.

Finally, Fig. 4 shows the classification of publications by topic, which were medical considerations (surgery recommendations, diagnosis, comorbidities, medical guidelines, dentistry considerations, among others), social and environmental considerations, general

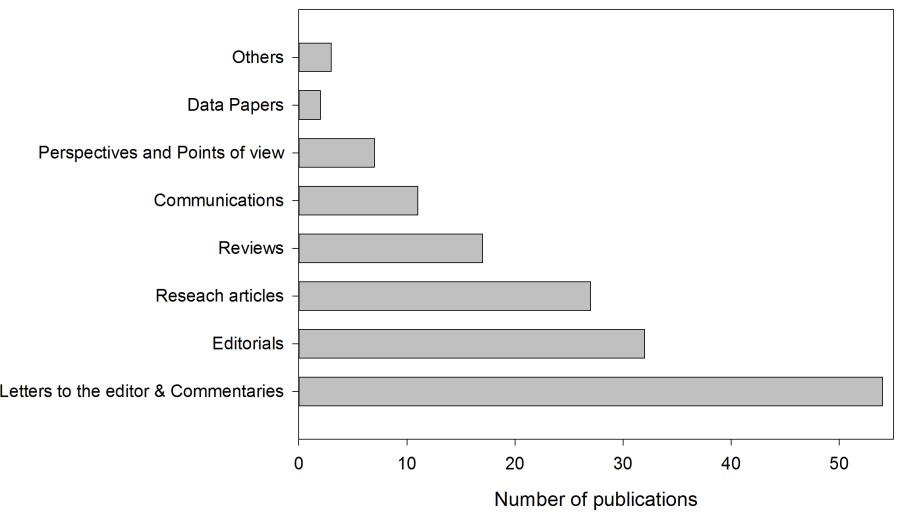

**Figure 2 Classification of publications by category.** Other publications refer to: consensus statement, contribution and technical note.

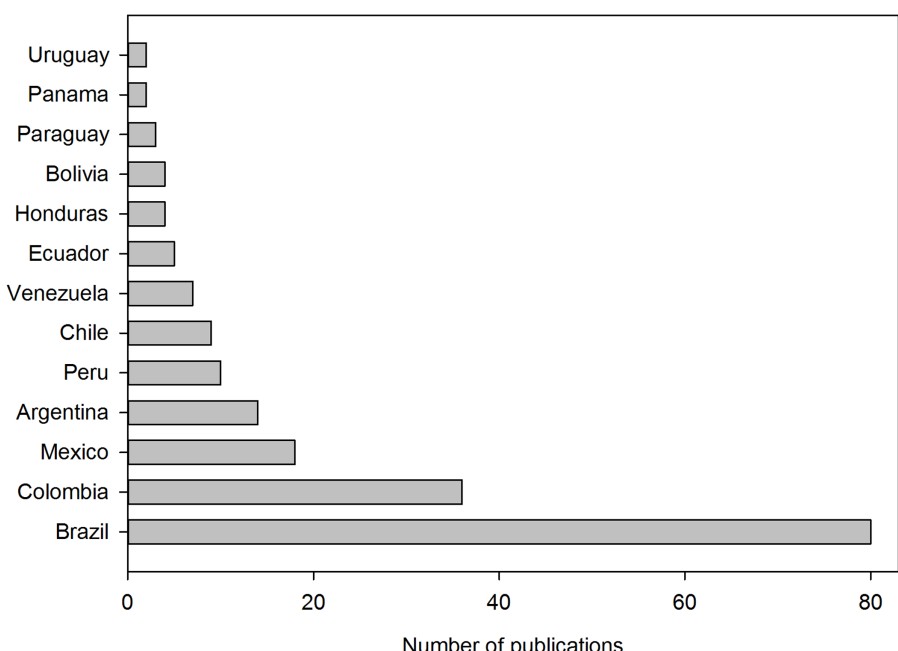

**Figure 3 Classification of publications according to the national affiliation.** Several authors from the same country in a publication were counted as one contribution to the country. Publications with authors from different countries were counted as one for each country. Latin American countries not shown had no publications until 10 April 2020.

aspects (zoonotic links, spreading, origin, disease, surveillance, among others), epidemiological analyses, bioinformatics (molecular and phylogenetic analyses, molecular simulations, genetic annotations, among others), mental health considerations, search for potential treatments, and meta-analyses. Excluding the general aspects, the remaining topics are shown in the following sections.

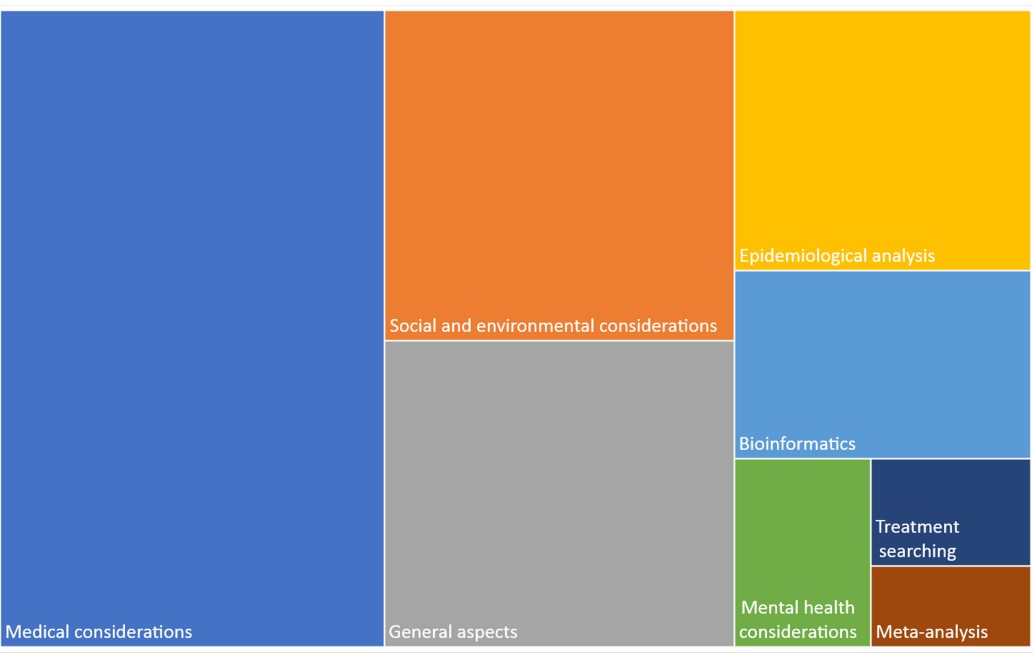

**Figure 4 Classification of publications by topic.** Medical considerations refer to surgery recommendations, diagnosis, comorbidities, medical guidelines, dentistry considerations, among others. The general aspects include zoonotic links, spreading, origin, disease, surveillance, among others. Bioinformatics refers to molecular and phylogenetic analyses, molecular simulations, genetic annotations, among others.

## PHYLOGENETIC AND MOLECULAR UNDERSTANDING

The researcher with the highest number of research articles was Ph.D. Marta Giovanetti, who has contributed to SARS-CoV-2 understanding through bioinformatic analyses (*Angeletti et al., 2020*; *Benvenuto et al., 2020a*, *2020b*, *2020c*, *2020d*; *Cleemput et al., 2020*; *Giovanetti et al., 2020a*, *2020b*). Her first research focused on a phylogenetic analysis of SARS-CoV-2, finding that among the viruses compared, this virus is closely related to bat-SL-CoVZXC21 (GenBank ID MG772934.1), while the least related is MERS (*Benvenuto et al., 2020b*). However, *Cárdenas-Conejo et al. (2020)* proposed that SARS-CoV-2 has a closer relation to bat-SL-CoV-RaTG13 (GenBank ID MN996532.1), then, authors suggested first that SARS-CoV-2 is unlikely to come directly from pangolin viruses, and second, that if SARS-CoV-2 has a recombinant origin, this recombination did not happen in ORF1ab. Nevertheless, both *Benvenuto et al. (2020b)* and *Cárdenas-Conejo et al. (2020)* concluded that the novel virus could come from a bat SARS-like coronavirus isolate, which is in agreement with reports from the GISAID database (*Hadfield et al., 2018*; *Sagulenko, Puller & Neher, 2018*).

Subsequent studies of the Giovanetti group found differences in the superficial spike protein S of SARS-CoV-2 through structural analyses, which could give a higher ability to infect humans when compared to other coronaviruses (*Benvenuto et al., 2020b*). This ability could be attributed to two mutations found in the non-structural protein 2 (nsp2) and nsp3, both originating from a possible positive pressure (*Angeletti et al., 2020*).

Similarly, evaluating in silico molecular interactions between the human angiotensin-converting enzyme 2 (ACE2) receptor and the spike protein of some coronaviruses, *Ortega et al. (2020a)* found that SARS-CoV-2 has some modified residues. Such residues could improve the recognition and interaction with the ACE2 receptor, providing SARS-CoV-2 with a higher infectiousness, which is in agreement with another study published simultaneously (*Andersen et al., 2020*). Likewise, performing in silico molecular interactions, *Ortega et al. (2020b)* evaluated the interaction between the protease of SARS-CoV-2 and some protease inhibitors as a strategy to control COVID-19 infection. The most energetic interactions predicted were using Saquinavir, Lopinavir, and Tipranavir, which are treatments for HIV patients; however, experimentation is required to validate these simulations.

In a later study, the Giovanetti group analyzed SARS-CoV-2 mutations through time, finding two variations located in nsp6 and ORF10, which could be caused by a positive selective pressure, leading to a lower protein structure stability and possibly (awaiting for evidence) a higher virulence (*Benvenuto et al., 2020a*). *Cárdenas-Conejo et al. (2020)* similarly detected variations in nsp6 and eight deleted amino acids in nsp1 from some Japanese virus strains. Although these in silico studies are a first approach and require experimental validation (*Ciccozzi et al., 2020*), they could also be a first step to aid in identifying treatments or vaccines.

Lastly, Giovanetti simultaneously contributed to another research group to develop and validate an open-access tool, called the Genome Detective Coronavirus Typing Tool, which analyzes SARS-CoV-2 genomes to generate new knowledge of COVID-19 outbreak (*Cleemput et al., 2020*).

Concerning the sequencing of SARS-CoV-2 genomes to understand this novel coronavirus, some Latin American researchers have contributed to the publication of sequences of isolated strains from countries such as Chile (*Castillo et al., 2020*) or Nepal (*Sah et al., 2020*). Researchers from other countries have also sequenced the genomes of strains from Argentina, Brazil, Chile, Colombia, Ecuador, Mexico, Panama, Peru and Uruguay, totaling 98 genome sequences until 20 April 2020 (see File S2 for detailed information of all sequences). The GISAID database has these 98 sequences collected along with 10,380 others, meaning that Latin American contribution is near 0.94% (*Hadfield et al., 2018*; *Sagulenko, Puller & Neher, 2018*). Table 2 summarizes this information by country, showing that Brazil and Mexico have the highest number of sequenced genomes, 52 and 17, respectively. Figure 5 presents some of the Latin American SARS-CoV-2 strains in the phylogenetic tree, evidencing the high heterogeneity among them because they belong to different clades.

# MEDICAL CONTRIBUTIONS

## Epidemiological analyses

Studies to track first cases in different countries have been performed, such as the case of tracing the first cases of COVID-19 in countries like Italy (*Giovanetti et al., 2020b*) and Chile (*Castillo et al., 2020*) using phylogenetic analyses. Both studies found that the first cases came from China and Europe since S and G variants of SARS-CoV-2 were detected.

**Table 2 Summary of genome sequences of SARS-CoV-2 strains isolated in Latin America and collected in GISAID database.** Information updated on 20 April 2020 (*Hadfield et al., 2018*; *Sagulenko, Puller & Neher, 2018*).

| Country | Submitting lab | Location | Total |
|---|---|---|---|
| Argentina | Instituto Nacional Enfermedades Infecciosas C.G.Malbran | Argentina | 3 |
| Brazil | Bioinformatics Laboratory—LNCC | Goiais | 1 |
| | | Minas Gerais | 5 |
| | | Rio de Janeiro | 7 |
| | | Rio Grande do Sul | 1 |
| | | São Paulo | 4 |
| | Instituto Adolfo Lutz, Interdiciplinary Procedures Center, Strategic Laboratory | Brasilia | 1 |
| | | Sao Paulo | 2 |
| | | Sao Paulo | 11 |
| | Instituto Oswaldo Cruz FIOCRUZ - Laboratory of Respiratory Viruses and Measles (LVRS) | Maceio | 1 |
| | | Feira de Santana | 2 |
| | | Brasilia | 5 |
| | | Vila Velha | 1 |
| | | Niteroi | 1 |
| | | Rio de Janeiro | 6 |
| | | Florianopolis | 1 |
| | | Joinville | 1 |
| | Laboratorio de Ecologia de Doencas Transmissiveis na Amazonia, Instituto Leonidas e Maria Deane—Fiocruz Amazonia | Manaus | 1 |
| | Laboratory of Virology | Brasilia | 1 |
| Chile | Instituto de Salud Publica de Chile | Santiago | 2 |
| | | Talca | 2 |
| | MSHS Pathogen Surveillance Program | Santiago | 3 |
| Colombia | Instituto Nacional de Salud Universidad Cooperativa de Colombia Instituto Alexander von Humboldt Imperial College-London London School of Hygiene & Tropical Medicine | Antioquia | 1 |
| | | Bogota | 1 |
| Ecuador | Institute of Microbiology, Universidad San Francisco de Quito | Pichincha | 3 |
| | | Quito | 1 |
| Mexico | Instituto de Diagnostico y Referencia Epidemiologicos (INDRE) | Chiapas | 1 |
| | | Estado de Mexico | 1 |
| | | Mexico City | 2 |
| | | Puebla | 1 |
| | | Queretaro | 1 |
| | Instituto Nacional de Ciencias Medicas y Nutricion Salvador Zubiran | Mexico City | 5 |
| | Instituto Nacional de Enfermedades Respiratorias | Mexico City | 5 |
| | Laboratorio Central de Epidemiología-DLVIE / Laboratorio de Secuenciación-Centro de Instrumentos. Instituto Mexicano del Seguro Social | Chihuahua | 1 |
| Nepal[a] | The University of Hong Kong | Kathmandu | 1 |
| Panama | Gorgas Memorial Institute for Health Studies | Panama City | 1 |
| Peru | Laboratorio de Referencia Nacional de Biotecnologia y Biologia Molecular. Instituto Nacional de Salud Peru | Lima | 2 |
| Uruguay | Microbial Genomics Laboratory, Institut Pasteur Montevideo | Montevideo | 9 |
| Total general | | | 98 |

**Note:**
[a] Nepal was included due to researchers with Colombian and Honduran affiliations contributed in its genome sequencing (*Sah et al., 2020*).
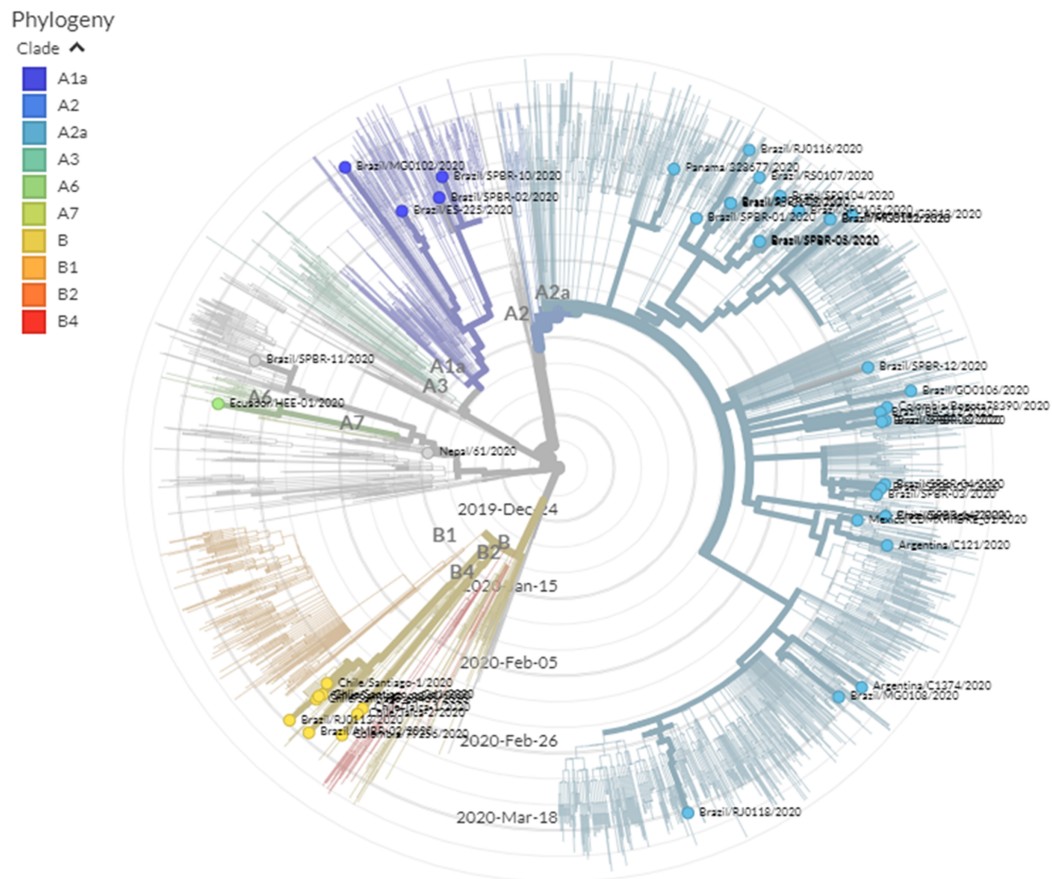

**Figure 5  Phylogenetic location of genome sequences of some strains isolated in Latin America in the SARS-CoV-2 phylogenetic tree.** The phylogenetic tree was generated and adapted from the GISAID database (*Hadfield et al., 2018*; *Sagulenko, Puller & Neher, 2018*). The tree was retrieved on 20 April 2020.                                                                                                      

In the case of Italy, a reentry from Germany was detected (*Giovanetti et al., 2020a*). Tracing of imported cases to other Latin American countries has also been performed using strategies such as data recovery of international flights from the most affected countries to Brazil (*Candido et al., 2020*), or recompilation of epidemiological data from hospitals in Bolivia (*Escalera-Antezana et al., 2020*). Both studies concluded that the first imported cases came from Europe, specifically from Italy and Spain. Moreover, other bioinformatics tools have been used to contribute to epidemiological understanding. For example, a Bayesian phylogeographic reconstruction suggested that Wuhan was effectively the outbreak epicenter on 25 November 2019, spreading later to other Chinese regions (*Benvenuto et al., 2020c*).

Additionally, the geo-positioning of some cases and the heterogeneity of the outbreak progress among countries, with demographic similarities, have been reported (*De Figueiredo et al., 2020*; *Xu et al., 2020*). Other epidemiological studies have been done to predict the COVID-19 spreading (*Benvenuto et al., 2020d*; *Córdova-Lepe, Gutiérrez-Aguilar & Gutiérrez-Jara, 2020*; *González-Jaramillo et al., 2020*; *Manrique-Abril et al., 2020*), which concluded that containment strategies are required to avoid overspreading in

countries like Chile and Colombia. Similarly, *Kraemer et al. (2020)* analyzed the effect of mobility in China before and after the sanitary containment of Wuhan in the COVID-19 spreading in this country; they demonstrated that decreasing mobility favors the reduction of COVID-19 spreading.

## Meta-analyses performed

Latin American researchers have also published meta-analyses of clinical, laboratory and image data from reported COVID-19 cases (*Do Nascimento et al., 2020*; *Rodriguez-Morales et al., 2020b*). On the one hand, the meta-analysis submitted by *Rodriguez-Morales et al. (2020b)* on 29 February 2020, used 19 articles with 2,874 patients for quantitative analyses. On the other hand, the meta-analysis of *Do Nascimento et al. (2020)*, submitted 6 days later, employed 61 studies with 59,254 patients. Both revisions included the most common symptoms as well as typical abnormalities observed in chest radiographs and computed tomographies, which were like other respiratory illnesses and viral pneumonia, making it difficult to distinguish from them (*Do Nascimento et al., 2020*). Oxygen support in critical patients was also analyzed, where *Do Nascimento et al. (2020)* suggested that excluding the non-invasive ventilation usage since no evidence supports its benefits.

Regarding lethality, the most affected population (81%) was the older group (60 years or more), but additional information is required to understand the COVID-19 impact on other continents because most of the data used in these meta-analyses came from China. Therefore, meta-analyses using Latin American cases would also be ideal for determining how COVID-19 could affect this region, which has some differences, such as lower average age or higher exposure to respiratory infections than other regions like Europe (*Amariles et al., 2020a*). A third meta-analysis was found, which suggested that it could be possible to predict if a patient with COVID-19 can present complications. *Lagunas-Rangel (2020)* suggested that complications are related to high levels of neutrophil-to-lymphocyte ratio (NLR) and low levels of lymphocyte-to-C-reactive protein ratio (LCR).

## Search for potential treatments

Concerning treatments to respiratory issues, *Khoury et al. (2020)* revised respiratory therapies for COVID-19 patients using cell-based treatments such as mesenchymal stem cells (MSCs), derivatives, or other cells, which have shown positive results in pre-clinical models of influenza. Conversely, they argued that there were few coronavirus studies, such as the case of seven COVID-19 patients in Beijing treated with MSCs, who showed apparent improvements up to 4 days after treatment, but lacked detailed information (*Leng et al., 2020*). Therefore, after a systematic search in databases, *Khoury et al. (2020)* found 27 ongoing treatments with 1,287 patients. The authors also highlighted the importance of following ethical protocols for these types of treatments.

Besides the previously mentioned review, since currently there are no approved treatments or vaccines for COVID-19, other revisions have summarized the development of clinical trials for COVID-19 treatments. This is the case of the search made by *Rosa & Santos (2020)*, who looked for ongoing clinical trials in Clinicaltrials.gov. The authors

used some constraints in their search, such as low cost, reduced time to reach markets, existing pharmaceutical supplies, or the possibility to combine with other drugs. In total, 24 clinical trials were found, most of them in clinical phases 2, 3, or 4, with a scheduled end in 2020. These ongoing clinical trials are using chloroquine, hydroxychloroquine, human immunoglobulin, remdesivir, arbidol, lopinavir, ritonavir, oseltamivir, darunavir, cobicistat, interferons, carrimycin, danoprevir, xiyanping, favipiravir, thalidomide, vitamin C, methylprednisolone, pirfenidone, bromhexie, bevacizumab, fingolimod, and traditional Chinese medicines (TCM).

Similarly, *Serafin et al. (2020)* searched for new treatments for coronaviruses like SARS-CoV, SARS-CoV-2 and HCoV-OC43 in PubMed, SCOPUS, and Web of Science databases. The drugs found as SARS-CoV-2 treatments are captopril, chloroquine, clomipramine, disulfiram, enalapril, hydroxychloroquine, mefloquine, metformin, nitazoxanide, remdesivir and teicoplanin. This review highlighted positive results using chloroquine, hydroxychloroquine, teicoplanin and hydroxychloroquine in conjunction with azithromycin. Finally, *Rosales-Mendoza (2020)* suggested that a potential treatment with biotechnological origin could be produced using plant species as hosts.

## Medical considerations

Most medical articles studied populations at risk and comorbidities, also provided recommendations and guidelines for medical personnel. *Rascado Sedes et al. (2020)* developed a contingency plan that allows an optimal response of the intensive care units (ICU) to the pandemic. The plan considers possible scenarios, the need for human and technical resources, communication and information, optimized use of resources, and personal protective equipment (PPE).

Healthcare workers, emergency room physicians, anesthesiologists, dentists, ophthalmologists, head and neck surgeons, maxillofacial surgeons, and otolaryngologists are among the most vulnerable ones because they perform procedures that can aerosolize secretions (*Kowalski et al., 2020*). According to several studies, articles, and protocols, *Boccalatte et al. (2020)* summarized recommendations related to PPE, mandatory use of protective suits, head covers, eye protection, mask, gloves, and N95, FFP2, or PAPR masks, personnel training, and techniques or manoeuvers for different medical practices.

Concerning surgeries, although there is no information about to translate risks to the operating room team during such procedures to COVID-19 patients, the recommendation is to postpone them (*Cohen et al., 2020*; *Ducournau et al., 2020*). The situations in which a delay in the surgical procedure may affect the patient, such as some oncologic or organ transplant surgeries, surgery must be performed following strict preventive measures. In the case of unpostponable abdominal surgeries, a laparotomic operation with regional anesthesia should be preferred (*Carneiro et al., 2020*; *Cohen et al., 2020*; *Correia, Ramos & Bahten, 2020*; *Quintão et al., 2020*). Other articles summarized protocols and recommendations for hand (*Ducournau et al., 2020*), head and neck (*Kowalski et al., 2020*), and urologic surgeries (*Puliatti et al., 2020*).

Researchers have also performed literature reviews concerning populations with higher risk due to comorbidities. First, *Hussain, Bhowmik & Do Vale Moreira (2020)* focused on

patients with diabetes mellitus and COVID-19. They summarized the different determinants that associate this pathology with greater severity and death, as well as the importance of multidisciplinary medical management. The authors mentioned that COVID-19 could cause pancreatic damage, which could affect patients with diabetes. Likewise, the positive effects of hydroxychloroquine and chloroquine on diabetic patients were mentioned, such as reducing insulin degradation tending to normalize glucose levels. Therefore, in case hydroxychloroquine or chloroquine is administered, antidiabetic drug doses should be readjusted to avoid hypoglycemic events. The authors also mentioned the necessity to continue studying in these patients the chronic inflammation, immune response, coagulation activity and vascular permeability, as well as whether hyperglycemia or hypoglycemia can alter the virulence of SARS-CoV-2, or whether the virus itself interferes with insulin secretion or glycemic control. All this information will be needed to propose adequate clinical treatments.

Likewise, *Puliatti et al. (2020)* considered the effect of SARS-CoV-2 in different organs of the urinary tract. They highlighted that ACE2-positive cells (target of SARS-CoV-2 spike proteins) have been found in these organs, which could have a high risk of affectation, even leading to death, which explains the kidney damage experimented in some COVID-19 patients. Similarly, chronic hemodialysis patients are also at particular risk due to their immunosuppression status, advanced age, and comorbidities coexistence. *Vega-Vega et al. (2020)* summarized the recommendations for these patients proposed by three international organizations: Center for Disease Control and Prevention (CDC), the Spanish Society of Nephrology, and the Latin American Society of Nephrology and Hypertension, to which were added suggestions of some experts. Some recommendations are the proper definition of cases, guidelines for patients and family, scrutiny of suspicious COVID-19 cases and their management within hemodialysis units, PPE employment, and sanitation of surfaces and devices.

Another group at risk mentioned by researchers is related to critical COVID-19 patients who need assisted ventilation. Some authors recommend taking into account some considerations before using high-flow nasal oxygen therapy, non-invasive ventilation, or extracorporeal membrane oxygenation, the latter being a treatment that no all medical centers can afford. First, if the medical center has adequate protection levels for health workers from exhaled air dispersed, and second, the impact of treatment on acute respiratory distress syndrome (*Bartlett et al., 2020*; *Ñamendys-Silva, 2020a*, *2020b*). Similarly, *Chica-Meza et al. (2020)* summarized conventional and non-conventional respiratory therapies in critical COVID-19 patients, including recommendations and considerations. Finally, surveys have been performed to identify COVID-19 impacts in patients at risk, such as pediatric patients with cancer (*Hrusak et al., 2020*), but the information is still limited; however, the recommendation is to follow the same medical practices described previously.

## Mental health considerations

Mental health care must be considered, given that quarantine can cause boredom, loneliness, anger, anxiety, depression and stress. Patients and health workers should have

mental health services (*De Medeiros Carvalho et al., 2020b*; *Lima et al., 2020*). In especial when more people can be affected by mental issues during the outbreak than by the outbreak itself (*Ornell et al., 2020*), and previous outbreaks have caused post-traumatic stress disorder in health workers (*Torales et al., 2020*).

Concerning performed studies on mental issues, first, *Fonseca et al. (2020)* established recommendations for patients with schizophrenia. Among recommendations are an adequate identification of COVID-19 symptoms, prevention of worsening of psychiatric symptoms, relapses due to the closed environment, fear of disease and isolation, use of telemedicine, promoting adherence to antipsychotic medication regimens, reducing emotional distress, hygiene practices, and family support. Finally, *Carvalho, Pianowski & Gonçalves (2020a)*, using questionnaires to Brazilians, investigated whether extroverted and conscientious people are engaged with the containment measures implemented during the COVID-19 pandemic. They found that extroverted people seem to lack commitment to containment measures, while people with conscientiousness personality tend to follow recommendations. Therefore, at least in Brazil, strategies for extroverted people should be proposed to avoid they become transmission vectors. However, the idiosyncrasy is similar throughout Latin America; thus, these strategies should be required in the entire region.

## ADDITIONAL CONCERNS

### Social and environmental concerns

- Telemedicine should be considered as an alternative to patient attention to avoid COVID-19 spreading. However, this strategy is limited because of the scarce existence of telemedicine systems; thus, social media could be used (*Machado et al., 2020*; *Quintão et al., 2020*).

- *Elachola, Ebrahim & Gozzer (2020)* highlighted that there are no prevalence and effectiveness studies on using facemasks related to the COVID-19 outbreak. This information could be useful in providing recommendations about their usage. However, based on past outbreaks, the evidence did not show positive or negative effects when the population used facemasks (*Stern et al., 2020*).

- Since the first COVID-19 cases reported in Latin America, fake news and misinformation have increased, where even treatments with potential health affectations have been proposed, as in the case of using chloroform or ether as alleged COVID-19 treatments (*Lana et al., 2020*; *Martins & Santos, 2020*). Additionally, information should be verified, avoiding panic spreading, which has caused panic buying of supplies and medications (*Cuan-Baltazar et al., 2020*).

- The quick COVID-19 spreading in different regions and countries could cause that health workers to become overwhelmed, therefore training new personal, although challenging, is a necessity that could help to attend the current outbreak. Additionally, the pharmacy workforce should be prepared as they are the first ones to attend possible cases of infection (*Amariles et al., 2020b*; *Aruru, Truong & Clark, 2020*; *Haines et al., 2020*).

- Countries with low or middle incomes, such as Latin American countries, may have already saturated their health systems. Hence, the COVID-19 outbreak could oversaturate them (*Carneiro et al., 2020*), which can be exacerbated by the lack of government preparation and social policies, such as the Ecuadorian case (*Hallo, Rojas & Hallo, 2020*).

- Based on events that occurred with the "Diamond Princess" cruise, alternatives should be proposed to quarantine people from other cruises and illegal immigrants entering a country (*Sawano et al., 2020*).

- Due to the current outbreak, many activities have been interrupted, which joined the socioeconomic limitations of vulnerable groups, could lead to food insecurity in these populations, who even have no access to clean water (*Oliveira, Abranches & Lana, 2020*).

- Knowing that COVID-19 has a zoonotic origin, One Health approach has taken relevance, which seeks integrative studies where the health of humans, environment and animals are considered to understand the virus environment, allowing the prevention or mitigation of future outbreaks (*Bonilla-Aldana, Dhama & Rodriguez-Morales, 2020a*).

## Medical concerns

- Initial reports of some pathologies such as rheumatic diseases or neuromyelitis optica spectrum disorder (NMOSD) have shown that they do not increase the risk of complications like other comorbidities. However, further studies are needed (*Carnero Contentti & Correa, 2020*; *Figueroa-Parra et al., 2020*).

- Angiotensin II receptor blockers (ATII-RB) are hypertensive drugs that increase ACE2 expression; therefore, there is a possibility that they can favor the internalization of SARS-CoV-2 within the cell; thus, further studies are required. However, suspending ATII-RB therapy may cause even higher affectations than COVID-19 itself; the risk-benefit ratio should be evaluated. In case doctors consider suspending it, there are other options like thiazide diuretics drugs (*Gracia-Ramos, 2020*; *Marin, 2020*).

- As previously mentioned, chloroquine and hydroxychloroquine are in vitro inhibitors of SARS-CoV-2 infection. However, concerns are focused on whether these medications could decrease viral load or prevent infection, clinical disease, clinical severity, or even death. Other factors, such as side effects, should also be considered (*Kim et al., 2020*; *Monteiro et al., 2020*; *Picot et al., 2020*).

- COVID-19 mainly affects the respiratory system, but it can also alter the central nervous system. Hence, possible affectations in the respiratory center could exacerbate respiratory distress caused by pulmonary affectation (*Conde et al., 2020*).

- Tropical countries affected by dengue could face two outbreaks at the same time, dengue and COVID-19, which could affect the population, even coinfecting some patients simultaneously. Both outbreaks will require the intensive attention of health systems to avoid a crossed affectation between them, which can be challenging and overwhelming

for the health systems (*Lorenz, Azevedo & Chiaravalloti-Neto, 2020*; *Navarro et al., 2020*; *Rodriguez-Morales, Sah & Paniz-Mondolfi, 2020*; *Saavedra-Velasco et al., 2020*).

- As SARS-CoV-2 can remain in saliva, oral health professionals require research focused on the influence of COVID-19 in their activities to take appropriate measures. However, the recommendation is to stop dental treatments (*Martelli-Júnior et al., 2020*; *Napimoga & Freitas, 2020*; *Sabino-Silva, Jardim & Siqueira, 2020*).

- Strategies during intubation are not the only important ones to avoid COVID-19 spreading in health workers during medical procedures; strategies for extubation are also required (*Trujillo, 2020*).

- There is no evidence that immunosuppressant treatments could decrease or increase the risk of severe COVID-19 infection; therefore, further investigation is recommended. In case treatments are suspended, factors such as potential issues on patients should be considered (*Carnero Contentti & Correa, 2020*).

- There is limited evidence of COVID-19 effect on pregnant women; hence, cases of pregnant women with COVID-19 should be studied to understand the clinical impact of the infection (*Zambrano et al., 2020*).

## CONCLUSIONS

Although our purpose was to give visibility to the contribution of Latin American researchers in the knowledge generation related to the COVID-19 outbreak, this review has two drawbacks. The first is the continuous availability of new publications; therefore, an observation window was employed. Second, several Latin American researchers are currently working on other continents without a Latin American affiliation, making them impossible to track. However, after this literature review, we were able to evidence the active participation of Latin American researchers in different subjects, whether as members of national, regional (LANCOVID-19), or even international research groups. Concerning our findings, the publications evidenced that these research groups have advanced in molecular and medical subjects, mainly in genetic understanding, epidemiological behaviors, meta-analyses, interaction between COVID-19 and other pathologies, and recommendations to medical procedures. Finally, understanding that this health crisis requires the commitment of as many researchers as possible, our wish is that the contribution of Latin American researchers continues to grow. Some topics with regional and global interest for future studies include in silico analyses of potential treatments and their respective in vitro and in vivo validations, meta-analysis of Latin American patients, and epidemiological surveillances. Regarding medical considerations, a deeper understanding of the COVID-19 interaction with risk comorbidities is needed to propose adequate clinical treatments. The same applies to unexplored/underexplored physical and mental pathologies, such as dengue.

## ACKNOWLEDGEMENTS

This review is in memory of all deceased by the COVID-19 outbreak.

### Funding
The authors received no funding for this work.

### Competing Interests
The authors declare that they have no competing interests.

### Author Contributions
- Karen Y. Fiesco-Sepúlveda conceived and designed the experiments, performed the experiments, analyzed the data, prepared figures and/or tables, authored or reviewed drafts of the paper, and approved the final draft.
- Luis Miguel Serrano-Bermúdez conceived and designed the experiments, performed the experiments, analyzed the data, prepared figures and/or tables, authored or reviewed drafts of the paper, and approved the final draft.

### Data Availability
Raw data are available in the Supplemental Files.

### Supplemental Information
Supplemental information for this article can be found online at http://dx.doi.org/10.7717/peerj.9332#supplemental-information.

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
