# Peer review of "Contributions of Latin American researchers in the understanding of the novel coronavirus outbreak: a literature review"

_PeerJ, doi:10.7717/peerj.9332_

## Round 0.1 · original submission · Major Revisions

I am personally interested in your study (since I work in Mexico) and will handle your submission. Before I send your manuscript to review, please
1) send your document to an English editing service or at least perform a grammar check (e.g., using Grammarly) to minimize errors in writing. Otherwise, the publication of your paper might be troubled or rejected for language reasons.

2) include information about the % contribution of Latinamerican (LA) institutions, compared to China, Europe, and the USA (maybe a global comparison would make sense?). This data would help to estimate the approx productivity of the region.

3) you included Nepal, because of a report of an LA author. However, I recommend focusing on the affiliations/ institutions rather than the nationality of individual scientists. E.g., in my institution, about 30% of the researchers are from abroad, and nobody would count our articles e.g., for Germany, Netherlands, etc.

---

## Round 0.2 · Minor Revisions

All three reviewers agree on the importance of our paper. I recommend that you emphasize the relevance to review the contribution of a certain region (e.g. could your findings be related to different cultures/ priorities/ funding situations?). Could you present any recommendations/ consequences/ perspectives, etc.?

·

Basic reporting

This a very interesting and timely report describing the research in a region, now affected by the COVID-19 significantly.

Experimental design

Good design.

Validity of the findings

Descriptive, valid findings.

Additional comments

Good initiative. My only recommendation is to update as much as possible, as this topic change daily.

·

Basic reporting

This paper employs clear, unambiguous and professional scientific writing throughout the manuscript. The English is generally excellent. The entire paper was clearly laid out and easy to follow. One thing the manuscript's paper asks is "What are the contributions of Latin American researchers in understanding the nCoV outbreak?" It would be worth considering adding this somewhere in the early part of the manuscript in a style befitting the excellent standard of English employed in the paper.

Throughout the manuscript, there is a necessary use of abbreviations, acronyms, author names necessitated by the language of the pandemic as well as the style sheet used by PeerJ—and yet the text does not feel weighted down. Figures and tables are cited within the text, and readers are referred to the back of the paper or appendices where they can be viewed. Overall, the paper reaches a high professional standard in structure. Only a few quibbles around the use of language and expression, which I will leave to later in this section.

A literature review of this type is of considerable interest to scientists within the countries examined, and by other countries. A review and characterization of the medical literature of COVID-19 is indicative of clinical, scientific and research activities in a region affected by the pandemic. While not a systematic exploration, this paper is an important type of bibliographic analysis. Another strength of the manuscript is how it teases out national perspectives where there is interdisciplinarity on research teams, and when cross comparisons with those countries affected by the pandemic in similar / different ways. This will be of value to research funding bodies, policy makers and researchers in other fields.

Experimental design

The study design is stated clearly in the title of the manuscript, and elucidated in the study methods. At times, I was looking for a bit more clarity which I will describe below.

A literature review whose methods are explicitly stated can be very effective in characterizing a corpus of literature. While not a systematic review, as long as a search is well-articulated and performed, and a system is described as to how the authors found and evaluated papers, this type of review can be useful. To orient readers from the outset as to your aims and purpose in the manuscript, I would recommend adding a sentence or two such as, "The aim of this paper is to do X" and "our method is a literature review". Do it as early as sentence #2. You describe your aims at lines 23-29, but it feels as though it would make sense earlier, "right off the top", and then think about deleting the sentence comprised by lines 20-22 (or move elsewhere).

With respect to the "Introduction", this section is sufficiently detailed and provides good background to the emergence of the Novel Coronavirus worldwide. Nearer the section's end, at lines 61-65, the purpose of the review is stated finally. The English could read a little more clearly; for example, say "literature survey" or "survey of the literature") not just survey) and say "submitted by these types of international research groups".

Search strategy 2.1 is appropriately reported but could be clearer and made more transparent. An overall statement should be made about why you selected those search engines, websites and biomedical databases. Was it because they were "openly available" as in PubMed? Easy and simple to use? Also, indicate the interfaces used if relevant. The keywords reported in your search strings as reported (you call them a "search equation" which is a phrase I have never heard before) are OK but some major omissions are evident.

For example, re: SARS-CoV-2 and COVID-19, why didn't you use the keyword phrase "novel coronavirus" OR "novel coronavirus disease*"? With respect to your geographic / Latin American countries filter, it is minimal and not sufficiently sensitive to capture all papers. As a result, some statement around the lack of consulting a search librarian expert, or doing a rather simplistic search in your paper, should be made. It is not clear to me whether a better search would have resulted in the retrieval of a more representative set of papers or not, but it's worth saying that it might be a limitation.

Sections 2.2--23 clearly state what the authors did in terms of data analysis but it was not clear how you selected the various section headings "Bibliometric analysis” and the following “Phylogenetic and molecular understanding” and “medical contributions” sections. Does this come out of a system of classifying papers? Or themes that emerged? It would be useful to have a sentence about how you came to those categories.

Validity of the findings

The validity of the findings in a literature review is contingent on a close reading of the papers, and a process chosen to characterize the papers. Again, it's not a systematic or even a rapid review. However, part of a literature review or survey of papers is to describe the papers retrieved in sufficient detail to know what they discuss. I feel that this is the leanest part of the paper. You cover other aspects well which are directly related to the aims of the paper, such as "Countries of authors" and affiliations (submitting lab at Table 2) which are sufficently described and illustrated.

Figure 2 is welcome. It would be interesting to see a range of topics "at a glance" as you have elucidated in the text of the paper but expressed as a table or even a word cloud.

The limitations of the search should be added to the conclusion.

Additional comments

This is a good paper, congratulations. I enjoyed reading it and hope you can provide periodic updates.

Reviewer 3 ·

Basic reporting

This review considers the contribution of Latin American research concerning Covid-19, which sets it apart from other reviews on this disease. Why is it important to consider the regional contributions to new science on this subject and does this contribute to understanding the unique aspect of every society when dealing with this pandemic? This should be covered in your introduction and conclusion sections.
The English language in this manuscript should be reviewed. There are many instances where the English language could be polished and I have indicated some of them below.
In general, when describing results reported by other studies, the past tense should be used.
On your abstract: “we searched in different databases, such as Scopus…” could be written as “we searched different databases, such as Scopus…”. “Finding that its contribution was 2.7…” would be clearer if written as “finding that the contribution of this continent was 2.7…”. Also, it is not clear from this sentence what this 2.7% represents. Finally, “visibility to the region contributions” could use an apostrophe (region’s). This is also true for line 61.
Line 21: on the first time the word “national” is mentioned, it could be deleted. Or maybe this affiliation could be better explained: are these 1797 affiliations or 1797 mentions to Latin American countries in the affiliations section?
On the introduction, the number 170000 could be written as 170,000. The same applies to 100000 and 5500.
Line 39: “as shown in table 1”
Line 43: “Coronaviridae, has a zoonotic origin, and can remain on some surfaces for considerable periods.”
Lines 45-47: this sentence needs rephrasing.
Line 81: “due to” should be replaced by “because”.
Line 101: “manually excluded eight additional publications due to affiliations from New Mexico…”
Line 124: “in consulted databases” can be deleted.
Line 127: delete “the.”
Line 130: The expression “science gap” is not very clear. Is it a gap in science funding, technology, facilities?
Line 144: “second, that if…”
Line 150: “a higher ability to infect humans when compared to other coronaviruses”
Line 157: “receptor, providing SARS-Cov-2 with a higher infectiousness”
Line 169: delete “While”
Lines 182-185: Please rephrase these sentences with active voice (table 2 summarizes…).
Line 202: “the geo-positioning of some cases and the heterogeneity of the outbreak progress among countries have been reported”
Line 212: “Meta-analyses and the search for potential treatments”
Line 221: “Oxygen support in critical patients…”
Line 229: “Concerning respiratory viruses”
Line 252: “use” should be deleted
Line 262: delete “regarding”
Line 303: “take into account”
Line 205: “the latter being a treatment that not all medical centres”
Line 309: “Finally, mental health care”

Experimental design

Line 83: Were there any articles with Latin American affiliation and no Latin American authors? Do you believe these articles should be completely excluded? Although they do not contain Latin American researchers, if there are any, you could mention them in your analysis since they were at least partially conducted in Latin American institutes.
Line 111: to avoid confusion, you could explain what the national affiliations correspond to. If one paper has authors affiliated to two different institutes of the same country, does this count as one national affiliation or two?
Lines 166-168: this information should be moved to the next paragraph.
Lines 226-228: this sentence could be rephrased (high levels of NLR and low levels of LCR). Was this confirmed or is this an initial hypothesis?
Lines 232-234: this study should be cited.

Validity of the findings

Reviews should be very cautious when reporting results from other studies; wording should be careful so as to not twist other studies’ results or imply different meanings.
I understand that your review considers Latin American contributions to understanding the epidemic, but the Latin American contribution should be compared with information published worldwide especially when there is a lack of published information or contradictory studies.
Line 39: This sentence is not very clear. Did you mean that the amount of testing per million inhabitants is low because of indigenous populations, vulnerable groups, and Venezuelan refugees? I have checked your reference (Torres & Sacoto, 2020) and the article stated that this happened due to limited test availability, and case monitoring was challenging because of the lack of cellphone use among these populations. Regarding Oliveira et al. (2020), I have not found any information on this subject, please check this reference citation.
Lines 141-142: was the highest of what? Among the analyzed viruses? Which ones?
Line 224: how old is this group exactly? What do you mean with most of the data came from China? Is there a need for more studies? Please make this clear.
Line 287: which are these positive effects of hydroxychloroquine and chloroquine on diabetic patients? Is this regarding blood glucose levels or COVID-19? According to your reference, these medications affect glucose levels. This should be made clear.
Line 320: It is important to state in which country these questionnaires were applied, since cultural and social aspects could also be involved in this matter, and please clearly state that their results represent that population and could not necessarily be the same for all countries.
Line 331: It is important to clarify if this lack of information about the use of face masks is observed only in studies published by Latin American authors or if this is a global issue.
Line 367: I believe the effects of angiotensin receptor blockers on Covid-19 infection have not yet been clearly established (see Reynolds et al., 2020, doi: 10.1056/NEJMoa2008975), therefore this topic could either relate the Latin American articles with other papers or clarify that this is a hypothesis.

---

## Round 0.3 · accepted · Accept

Although the paper is acceptable, a thorough review by a native-equivalent speaker/editorial service is recommended. PeerJ does not provide this as part of their production process. E.g., the title should be "Contributions of Latin American researchers in the understanding of the novel coronavirus outbreak: A literature review" ('of' is missing).

The first sentence of the Abstract can be deleted. You can start directly with "The world is facing..". Several words are superfluous, e.g., in the Abstract "Therefore, ... For instance, ... ".

You should perform this optimization of the text in the interest of your readers and the impact of your article.